# Current Status and Perspectives of Dual-Targeting Chimeric Antigen Receptor T-Cell Therapy for the Treatment of Hematological Malignancies

**DOI:** 10.3390/cancers14133230

**Published:** 2022-06-30

**Authors:** Bailu Xie, Zhengdong Li, Jianfeng Zhou, Wen Wang

**Affiliations:** 1IASO Biotherapeutics Co., Ltd., No.77-78, Lane 887, Zuchongzhi Rd, Pudong, Shanghai 200120, China; bailu.xie@iasobio.com; 2Department of Breast Surgery, Shanghai First Maternity and Infant Hospital, School of Medicine, Tongji University, Shanghai 200020, China; lzd-shy@sohu.com; 3Department of Hematology, Tongji Hospital, Tongji Medical College, Huazhong University of Science and Technology, Wuhan 430030, China; jfzhou@tjh.tjmu.edu.cn; 4IASO Biotherapeutics Co., Ltd., San José, CA 95138, USA

**Keywords:** chimeric antigen receptor T cells, antigen escape, dual-targeting, bispecific CAR, hematological malignancies

## Abstract

**Simple Summary:**

Approved chimeric antigen receptor (CAR) T cells recognize and bind to only one tumor target (single-targeted CAR T cells, Si-CART) on cancer cells by the special receptor and followed with activation, thus removing cancers from patients. However, cancer cells can resist the treatment of Si-CART by hiding the single target to prevent the recognition and survive, causing recurrence of cancers in patients. Dual-targeting CAR T-cell therapy contains CAR T cells recognizing two targets on cancer cells and can overcome the resistence in cancers to Si-CART. We summarize the latest preclinical and clinical development of dual-targeting CAR T-cell therapies to provide perspectives for optimization and shed light on new hope for patients after the treatment of Si-CART.

**Abstract:**

Single-targeted chimeric antigen receptor (CAR) T cells tremendously improve outcomes for patients with relapsed/refractory hematological malignancies and are considered a breakthrough therapy. However, over half of treated patients experience relapse or refractory disease, with antigen escape being one of the main contributing mechanisms. Dual-targeting CAR T-cell therapy is being developed to minimize the risk of relapse or refractory disease. Preclinical and clinical data on five categories of dual-targeting CAR T-cell therapies and approximately fifty studies were summarized to offer insights and support the development of dual-targeting CAR T-cell therapy for hematological malignancies. The clinical efficacy (durability and survival) is validated and the safety profiles of dual-targeting CAR T-cell therapy are acceptable, although there is still room for improvement in the bispecific CAR structure. It is one of the best approaches to optimize the bispecific CAR structure by boosting T-cell transduction efficiency and leveraging evidence from preclinical activity and clinical efficacy.

## 1. Introduction

Single-targeted chimeric antigen receptor (Si-CAR) T cells (Si-CART) have tremendously improved outcomes for patients with relapsed hematological malignancies, such as acute lymphoblastic leukemia (ALL), non-Hodgkin lymphoma (NHL), and multiple myeloma (MM). CD19 and B-cell maturation antigen (BCMA) are the two most successful antigens for engineering Si-CART, with excellent response rates. The overall response rates (ORR) for the approved products targeting CD19, lisocabtagene maraleucel [1], axicabtagene ciloleucel [2], tisagenlecleucel [3], and brexucabtagene autoleucel [4], have reached more than 70%, while the ones targeting BCMA, idecabtagene vicleucel and ciltacabtagene autoleucel, have reached 73% and 97%, respectively [5,6]. However, at least 50% of patients still experience relapse or refractory disease after treatments with CD19 Si-CART [7], whilst approximately 60% of patients with heavy prior treatments did not remain progression-free 12 months after BCMA Si-CAR T-cell therapy [8]. This poses an unmet medical need for patients who did not gain long-term benefits from chimeric antigen receptor (CAR) T-cell therapy.

The main mechanisms for relapse after treatments with Si-CART are the restricted persistence of CAR T-cells (CAR-T), inhibition of CAR T-cell function, and antigen escape [9]. Antigen escape occurs when tumor cells evolve to express a low level of antigen to prevent the recognition of Si-CART, resulting in the failure of Si-CART to bind to the intended target [10]. To minimize the risk of relapse due to target escape, strategies that use CAR-T to recognize more than one tumor-related antigen in malignant cells are actively being explored in clinical trials [9,11,12,13]. Dual-targeting CAR T-cell therapy utilizes dual CAR strategies to identify two tumor-associated antigens in cancer. This can be achieved using two pooled Si-CAR T-cell products with different antigen-binding specificities or a single CAR T-cell product capable of targeting two different antigens [11]. We will refer to the latter, capable of targeting two tumor-associated antigens as bispecific CAR T-cell therapies (Bi-CART). These are usually generated by transduction with a bivalent vector or a bicistronic vector. In preclinical models and in clinical trials for hematological malignancies, dual antigens for dual-targeting CAR T-cell therapy have three combinations: CD19/CD20, CD19/CD22, and BCMA/other targets on plasma cells [12,13,14,15,16]. The cell types and expression patterns of CD19, CD20, CD22, and BCMA are well characterized [8,17,18,19]. Recently, the increasing use of Bi-CART in clinics and comprehensive clinical data on Bi-CART being disclosed make a thorough analysis of dual-targeting CAR T-cell therapy from bench to bedside possible. The current review aimed to demonstrate detailed data on the efficacy and safety of dual-targeting CAR T-cell therapy along with bispecific CAR (Bi-CAR) structure optimization to gather evidence for developing Bi-CAR T-cell (Bi-CAR-T) therapy.

## 2. Common Dual CAR Strategies

The main CAR structures used in dual-targeting CAR T-cell therapy are Si-CAR, bivalent tandem CAR, bivalent loop CAR, and bicistronic CAR. As illustrated in Figure 1, dual-targeting CAR T-cell therapy can be categorized into the following five dual CAR strategies:(i)Cocktail/sequential infusion of two separate Si-CAR T-cell products

Two individual Si-CAR T-cell products are first produced by transducing T cells with two different vectors separately. Next, two separate Si-CAR T-cell products are pooled together at a ratio of 1:1 to infuse on the same day. Alternatively, the two Si-CAR T-cell products are infused on consequent days (Figure 1A, Cocktail/sequential).

(ii)Heterogeneous cell products of Si-CART and Bi-CART resulted from co-transduction of two separate vectors

A dual CAR T-cell product can be produced by co-transduction of T cells with two separate vectors, each of which encoding one individual CAR structure. It contains two separate Si-CAR T-cells and one Bi-CAR T-cells in the final pooled product (Figure 1B, Co-transduction).

(iii)One Bi-CAR T-cell product with bicistronic CAR (Bicistronic Bi-CART)

One bicistronic vector is introduced into T cells to generate dual distinct CARs (Figure 1C, Bicistronic), resulting in one Bi-CAR T-cell product with two separate CARs with each antigen-binding domain.

(iv)One Bi-CAR T-cell product with bivalent tandem CAR (Tandem Bi-CART)

One bivalent vector is introduced into T cells to produce dual domains to bind two different antigens within one Bi-CAR. Bivalent CAR can be categorized into two different structures, tandem and loop, by placing the variable light chain (V_L_) and variable heavy chain (V_H_) of single-chain variable fragments (scFv) in a different order. The tandem structure is formed with V_L_-V_H_ of one scFv directly linked to the V_L_-V_H_ of the other scFv (Figure 1D, Bivalent/Tandem).

(v)One Bi-CAR T-cell product with bivalent loop CAR (Loop Bi-CART)

Similar to Tandem Bi-CART, one bivalent vector is introduced into T cells to produce dual domains to bind two different antigens within one Bi-CAR. The loop structure is formed with V_L_-V_H_ of one scFv separated by the V_L_-V_H_ of the other scFv (Figure 1E, Bivalent/Loop), which is different from bivalent tandem CAR.

Recently a growing number of dual CAR strategies have been proposed. Of note, two pooled Si-CAR T-cell products and the Bi-CAR T-cell product are sometimes referred to as OR-gate CAR-T or CAR-T using “OR” logic gate (activated by one antigen or the other on tumor cells) [20,21]. CAR-T using other logic gates [20], such as “AND” (only activated when recognizing both antigens on tumor cells) [22], “NOT” (inactivated when encountering one antigen on normal cells) [23], and “synNotch” (first primed and induced by one tumor-specific but heterogeneous antigen and then activated by one homogeneous antigen or the other homogeneous antigen) [24,25], are also being developed predominantly for solid tumors. Bi-CAR T cells with anti-epidermal growth factor receptor splice variant III or anti-myelin oligodendrocyte glycoprotein synNotch–anti-ephrin type-A receptor 2, and interleukin 13 receptor α2 tandem CAR displayed precise brain tumor control in a mouse model of glioblastoma [24]. Likewise, Bi-CAR T cells expressing alkaline phosphatase placental-like 2 synNotch CAR circuits with mesothelin CAR for mesothelioma or epidermal growth factor receptor 2 CAR for ovarian cancer exhibited higher efficacy than Si-CAR T-cells in preclinical studies [25]. Dual CAR strategies also include generation of CAR T-cells by transduction with a bicistronic vector or two separate vectors encoding CAR and inhibitory CAR structure [11]. Novel dual CAR strategies certainly help to enrich the armory against tumor cells.

The theoretical advantages and disadvantages of dual CAR strategies are summarized in Table 1. Comparison of transduction efficiencies and effects among different dual CAR strategies in vitro and in vivo can be found in Table 2. Considering investigation of Bi-CAR T-cells transduced with different constructs head to head in the clinical setting is not yet feasible, observation in preclinical studies might offer some insight for the optimization of the Bi-CAR structure. Unlike the cocktail/sequential infusion of manufactured Si-CAR T-cell products at our disposal, to bind two different antigens, the corresponding antigen-binding fragments need to be efficiently engineered into T cells to produce Bi-CAR structure on the T cell membrane effectively. This process includes designing suitable CAR constructs, generating vectors for the viral package, producing viruses for transduction, and establishing Bi-CAR T cells by viral transduction.

Some features of Bi-CAR construct are related to druggability. Transduction efficiency is a major issue that needs to be improved. The length of co-stimulatory domains and linkers for producing Bi-CAR construct requires a large-sized vector, causing difficulty in viral packaging [44,45] and reducing transduction efficiency [46]. Poor transduction efficiency leads to reduced Bi-CAR construct expression on the T cell membrane. A bicistronic vector needs a long sequence to encode two co-stimulatory domains for bicistronic CAR. By comparison, the bivalent vector is smaller than the bicistronic vector because only one co-stimulatory domain needs to be generated by the vector. Therefore, lower transduction efficiencies in bicistronic Bi-CAR T cells targeting BCMA/CS1 and CD5/CD7 were observed, compared to bivalent Bi-CAR T cells [41,42]. Similarly, the transduction efficiency in the CD19/CD22 Loop Bi-CAR T cells expressing LoopCAR6 with a longer linker was lower than the Loop Bi-CAR T cells expressing LoopCAR3 with a shorter linker [33]. However, several investigations showed that transduction with a bivalent vector is not always superior to transduction with a bicistronic vector in terms of transduction efficiency and ultimate activities of Bi-CAR T cells. The transduction rate in preclinical models using normal peripheral blood mononuclear cells (PBMC) can reach 60% in BCMA/G-protein-coupled receptor class 5 member D (GPRC5D) Bi-CAR T cells [39] and 72% in a CD138/CD38 Bi-CAR T cells [40], close to those in bivalent CAR T cells. The survival of mice treated with BCMA/GPRC5D Bicistronic Bi-CAR T cells was longer than that with Tandem Bi-CAR T cells in BCMA-GPRC5D+ models [39]. In contrast, BCMA/CS1 Tandem Bi-CAR T cells performed better than Bicistronic Bi-CAR T cells in terms of CAR surface expression, transduction efficiency, and CAR T cell proliferation, resulting in further in vivo studies on Tandem Bi-CAR T cells [42]. Whether those results are related to different sequences, manufacturing methods, or the status of T cells from different patients in individual studies remains to be determined. Non-viral transduction using transposons for producing Bi-CAR T cells is possible to increase transduction efficiencies owing to the capacity of transferring large gene constructs [47].

The transduction efficiency ranged from 7.4% to 28% in one CD19/CD20 Tandem Bi-CAR T-cell product [27] and 10.32% to 16.91% in another CD19/CD22 Tandem Bi-CAR T-cell product [29]. These transduction efficiencies were from cell products generated from apheresis in clinics and were lower than those reported (>50%) in most preclinical studies. The transduction rate for BCMA/CD38 Tandem Bi-CAR T cells from patients in the clinical study can drop to 12%, even though 59.4% was reported in the same Tandem Bi-CAR T cells from PBMC of healthy donors used in the preclinical models [38], indicating that transduction efficiency in preclinical models may not be able to predict outcomes in the clinical setting.

Besides the transduction efficiency, the spatial structure of two scFvs also affects the Bi-CAR T cell activities. A CD19/CD22 Loop Bi-CAR with membrane-proximal CD22 CAR was shown to be more effective than the Tandem Bi-CAR in eradicating tumor cells and prolonging survival in mouse models [33]. Furthermore, as demonstrated in studies on CD19/CD20 Bi-CAR T cells and CD19/CD22 Bi-CAR T cells, the shorter distance from scFv to the target on the cell membrane can lead to a higher activity of CAR T cells [21,33], supporting the development of a bivalent vector encoding short linker to connect two scFvs of Bi-CAR.

## 3. Clinical Efficacy and Safety

### 3.1. Clinical Efficacy of Dual-Targeting CAR T-Cell Therapy for Hematological Malignancies

Currently, investigations reveal that there are four main mechanisms responsible for relapse due to antigen escape: (i) receptor genetic mutations [9,10,48], (ii) cell lineage switch [9,10,48], (iii) epitope masking [9,10,48], and (iv) trogocytosis [49]. The loss of receptors on the membrane is attributed to CD19 mutants in exons 2–5 arising from DNA genetic alteration and alternative RNA splicing, which were detected in 19 patients in the clinics and prevented the recognition of CD19 Si-CAR T cells [50,51]. This mechanism led to the rationale for dual-targeting CAR T-cell therapy that could eliminate CD19-negative malignant B cells, which retain CD20 or CD22. Lineage switching, such as transformation from a lymphoblastic lineage to a myeloid lineage [52,53] or from chronic lymphoblastic leukemia to plasmablastic lymphoma [54], has been identified, resulting in the loss of CD19 and even other B-cell antigens, including CD20 and CD22 expression. To overcome this mechanism, Bi-CAR T cells, targeting unusual antigens other than B-cell antigens, needs to be explored during early discovery. Ruella et al. (2018) reported a rare case of epitope masking caused by unintentionally transducing B cells with CAR construct against CD19; the expression of CAR on the resulting CAR-transduced B cell leukemia cells (CARB) bound to the CD19 epitope of the same CARB, thus, blocking the binding of CD19 Si-CART to CARB [36]. This was caused by CAR T cell manufacturing [36], which cannot be solved by dual-targeting CAR T-cell therapy. In recent years, tumor cells are found to be able to transfer the target antigen to CAR T cells via trogocytosis, resulting in diminished antigen expression on tumor cells and fratricide of CAR T cells [49].

Published clinical data on relapse after CD19 Si-CAR-T therapy until 2018 were eloquently summarized by Majzner and Mackall [10]. In four trials, 37 of 220 patients with ALL experienced CD19-negative relapse after treatments with CD19 Si-CAR T-cell therapy [10], with median follow-up ranging from 12 [55], 13.1 [3], 22.6 [56], to 29 months [57], respectively. The level of CD19 expression in NHL after treatments of Si-CAR-T and dual-targeting CAR-T therapy has not been well summarized, possibly due to false negativity since tumor tissue heterogenicity or sampling that can lead to an unreliable conclusion. Some trials reported CD19 expression as negative or positive [4,28,58], while one trial reported percentages in which no specific number was interpreted as CD19-negative or CD19-low expression [27]. In a meta-analysis study on CD19 Si-CAR T-cell therapy, the median progression-free survival (PFS) of subjects with B cell malignancies was 7 months [59], whereas time to CD19-negative or CD19-low relapse has not been well analyzed. The time to CD19-negative or CD19-dim relapse was reported to be around 2–3 months in five patients, 4–6 months in four patients, 8–9 months in five patients, and 14 months in one patient [50,51]. Despite the difficulty in sampling in clinical trials, it may be of value to gather more data on the time to CD19-negative or CD19-low relapse to serve as a parameter for future investigation.

The advantage of dual-targeting CAR T-cell therapy over Si-CAR-T-cell therapy is its ability to decrease antigen escaping of tumor cells. Clinical studies of Si-CAR T-cell therapy have already shown >90% complete response (CR) [30,60], leaving little room for improvement in terms of the initial response to dual-targeting CAR T-cell therapy. Therefore, the expectation for dual-targeting CAR T-cell therapy is not only to improve the durability of the response but also to reinduce the response in patients who relapsed or were refractory after treatments with Si-CAR T-cell therapy. Table 3 and Table 4 provide data questioning whether dual-targeting CAR T-cell therapy can override Si-CAR T-cell therapy in durability and long-term clinical benefit, e.g., longer duration of response (DOR) and overall survival (OS). It seems that dual-targeting CAR T-cell therapy has demonstrated better DOR and OS than Si-CAR T-cell in a small number of studies. However, there were no head-to-head studies and, therefore, the conclusions should be interpreted with caution due to differences, such as disposition of patients and supportive care between studies. Similar results were found when comparing the data from different studies. In ALL, 6-month RFS and OS were similar between CD19 Si-CAR T-cell product tisagenlecleucel [3] and CD19/CD22 Bi-CAR T-cell therapy [61]. Likewise, the 12-month PFS for NHL patients was close among tisagenlecleucel in diffuse large B-cell lymphoma [58], brexucabtagene autoleucel in mantle-cell lymphoma [4], and CD19/CD20 Tandem Bi-CAR T-cell therapy in B-cell lymphoma [28]; however, the comparison should be viewed with caution among different clinical entities. In particular, in one trial with a head-to-head comparison of CD19 Si-CAR T cells with CD19/CD22 Bi-CAR T cells, the median leukemia-free survival (LFS) in patients without hematopoietic stem cell transplantation (HSCT) after CAR T-cell treatment was 2 months for CD19 Si-CAR T cell treatment, while LFS was 3 months for CD19/CD22 Bi-CAR T cell treatment, demonstrating a better DOR of Bi-CAR T-cell therapy [30].

Significant differences were observed during the comparison. For example, OS for ALL patients treated with CD19 Si-CAR T cells [3,30] was close to those with CD22 Si-CAR T cells [62], which was shorter than those treated with CD19/CD22 Bi-CART cells [30]. OS for ALL patients treated with cocktail infusion of CD19/CD22 Si-CAR T cells [16] was the longest among those treated with CD19 Si-CAR T cells, CD22 Si-CAR T cells and Bi-CAR T cells. In ALL, CD22 Si-CAR T cells performed poorer than CD19/CD22 Bi-CAR T cells with regard to the 6-month RFS. In NHL, the percentages of PFS and OS in a trial on CD19/CD20 Bi-CAR T-cell threapy [28] were higher than those in a trial with CD19 Si-CAR T-cell therapy [1], despite the ten-fold enrollment in the latter. Whether results from a small sample size can be reproduced in an expanded cohort with head-to-head comparision remains to be determined.
cancers-14-03230-t003_Table 3Table 3Comparison of dual-targeting CD19/CD22 CAR T-cell therapy with the respective Si-CAR T-cell therapy with respect to duration of response, survival, and expansion in ALL.Ref.: First AuthorTargetCAR StrategySample Size (CR Patients)DurabilityOS (mon and %)In Vivo ExpansionMaude [60]CD19One Si-CAR-T product30 (27 CR)NA78% (6-mon OS)Median C_max_: 39.8%C_max_: >5000 copies/μg gDNA (>15,000 copies/μg gDNA in 26 pts)Maude [3]CD19One Si-CAR-T product75 (61 CR)73% (6-mon RFS),50% (12-mon RFS)19.1 mon (median OS),90% (6-mon OS),76% (12-mon OS)Median Tmax: 10 daysC_max_: NAGrupp [63]CD19One Si-CAR-T product79 (65 CR)66% (18-mon PFS);Responses were ongoing in 29 pts (max DOR, 29 mon and ongoing)70% (18-mon OS)NA Shah [62]CD22One Si-CAR-T product56 (40 CR)31.6 mon (EFS),6 mon (RFS in CR),11 remain in remission with a median f/u of 9.7 mon13.4 mon (median OS)T_max_: days 14∼21Median C_max_: 77% CAR+T cells; 480.5 CAR-T/μL Wang [30]CD19One Si-CAR-T product35 (31 CR)∼2 mon (median LFS in 19 non-HSCT pts)∼12 mon (median OS in all pts)T_max_: day 10.5Median C_max_: 590.4 CAR-T/μLWang [30]CD19/CD22One Tandem Bi-CAR-T product15 (13 CR)∼3 mon (median LFS in 13 non-HSCT pts)∼21 mon (median OS in all pts)T_max_: day 9Median C_max_: 448.2 CAR-T/μLWang [16]CD19/CD22Cocktail/Sequential infusion of two Si-CAR-T products51 (48 CR)52.9% (12-mon PFS)13.6 mon (median PFS)62.8% (12-mon OS)31 mon (median OS)Median T_max_ and Mean/Median C_max_ NAPan [15]CD19/CD22Cocktail/Sequential infusion of two Si-CAR-T products20 (20 CR)79.5% (12-mon LFS)92.3% (12-mon OS)Median T_max_ and Mean/Median C_max_ NASchultz [64]CD19/CD22One Bivalent Bi-CAR-T product12 (10 CR)NA92% (9.5-mon median f/u)Median C_max_: 11.13% (Dose Level 1) and 29.1% (Dose Level 2) Dai [29]CD19/CD22One Tandem Bi-CAR-T product6 (6 CR)≥ 5 mon (RFS in 5 CR, 3 ongo-ing > 8 mon, 1 relapse after 3 mon)NAMedian T_max_ and Mean/Median C_max_ NAYang [35]CD19/CD22One Loop Bi-CAR-T product16 (>6/7 CR)3 mon (median observed time without relapse) NAMedian C_max_: 109,000 copies/μg gDNATang [61]CD19/CD22One Tandem Bi-CAR-T product22 (22 CR)76.9% (6-mon RFS),67.3% (12-mon RFS)94.4% (6-mon OS), 57.2% (12-mon OS)NASpiegel [34]CD19/CD22One Loop Bi-CAR-T product17 (15 CR)5.8 mon (PFS)11.8 mon (median OS)Median C_max_: 36 CAR-T/μL1794 copies/50 ng gDNAT_max_: days 10–14Cordoba [32]CD19/CD22One Bicistronic Bi-CAR-T product15 (13 CR)48% (6-mon EFS),32% (12-mon EFS)80% (6-mon OS),60% (12-mon OS)C_max_ > 30,000 copies/μg DNAMedian T_max_: 12 daysAbbreviations: ALL, acute lymphoblastic leukemia; C_max_, peak of CAR-T/Peak CAR; CR, complete response; EFS, event-free survival; f/u, follow-up; gDNA, genomic DNA; LFS, leukemia-free survival; mon, month(s); NA, not available; non-HSCT, no hematopoietic stem cell transplantation; OS, overall survival; PFS, progression-free survival; pts, patients; Ref., reference; RFS, relapse-free survival; T_max_, the median time to maximum expansion.
cancers-14-03230-t004_Table 4Table 4Comparison of dual-targeting CD19/CD20 or CD19/CD22 CAR T-cell therapy with the respective Si-CAR- T-cell therapy regarding duration and survival in NHL.Ref.: First AuthorTargetCAR StrategySample Size (CR Patients)DurabilityOS (mon and %)In Vivo ExpansionLocke [65]CD19One Si-CAR-T product73 ongoing CR at 12+monNAMedian T_max_ and Mean/Median C_max_ NALocke [66]CD19One Si-CAR-T product10811.1 mon (Median DOR),44% (12-mon PFS)59% (12-mon OS)Median T_max_ and Mean/Median C_max_ NASchuster [67,58]CD19One Si-CAR-T product93–99Median DOR NR (10 mon-NR),66% (12-mon PFS)49% (12-mon OS)Median T_max_ and Mean/Median C_max_ NAJacobson [68]CD19One Si-CAR-T product10965.6% (18-mon PFS)87.4% (18-mon OS)Median T_max:_ 9 daysMedian C_max_ NAAbramson [1]CD19One Si-CAR-T product2696.8 mon (PFS),51.4% (6-mon PFS),44.1% (12-mon PFS);Median DOR NR (8.6-NR)74.7% (6-mon OS), 57.9% (12-mon OS)Median T_max:_ 12days Median C_max_: 23,928.2 copies/μg gDNAWang [4]CD19One Si-CAR-T product6061% (12-mon PFS)83% (12-mon OS)Median T_max:_ 15 daysMedian C_max_ NAZhang [69]CD20One Si-CAR-T product11>6 mon (PFS), 1 CR for 27 monsNAMedian T_max:_ ∼28 daysMedian C_max_ NATong [28]CD19/CD20One Tandem Bi-CAR-T product2779% (6-mon PFS),64% (12-mon PFS)82% (6-mon OS),71% (12-mon OS)Mean C_max:_ 496 CAR-T/μLMedian T_max_: NAShah [27]CD19/CD20One Tandem Bi-CAR-T product2212 CR > 6 mon; 6 CR > 12 mon; 8 CR ongoingNAMedian T_max_ and Mean/Median C_max_ NATholouli [70]CD19/CD22One Bicistronic Bi-CAR-T product354 CR > 10 mon; 4 CR > 5 mon.NAMedian T_max_ and Mean/Median C_max_ NAWang [16]CD19/CD22Cocktail/Sequential infusion of two Si-CAR products369.9 mon (median PFS)50.0% (12-mon PFS)18.0 mon (median OS)55.3% (12-mon OS)Median T_max_ and Mean/Median C_max_ NAZhang [31]CD19/CD22One Loop Bi-CAR-T product3240.0% (12-mon PFS)66.7% (12-mon PFS in CR at 3 mon)63.3% (12-mon OS)100% (12-mon OS in CR at 3 mon)Median T_max:_ 12 daysGeometric mean C_max_: 286,294.4 copies/μg DNASpiegel [34]CD19/CD22One Loop Bi-CAR-T product213.2 mon (median PFS)22.5 mon (median OS)C_max:_ 36 CAR-T/μL1794 copies/50 ng gDNAT_max_: days 10–14Abbreviations: C_max_, peak of CAR-T/Peak CAR; CR, complete response; EFS, event-free survival; f/u, follow-up; FL follicular lymphoma; gDNA, genomic DNA; LFS, leukemia-free survival; mon, month(s); NA, not available; NR, not reached; non-HSCT, no hematopoietic stem cell transplantation; NHL, non-Hodgkin lymphoma; OS, overall survival; PFS, progression-free survival; pts, patients; Ref., reference; RFS, relapse-free survival; T_max_, the median time to maximum expansion.

Strategies involving CD19/CD20 or CD19/CD22 to design Bi-CAR T-cell therapy are based on the hypothesis that targeting CD20 or CD22 would benefit patients with a loss or reduction in CD19 to overcome antigen escape. After careful extraction of data from published clinical trials, details of CD19 expression and the related efficacy were not identified. Only patients with positive CD19 expression were enrolled in several studies, resulting in limited data concerning whether reinduction of CR can be achieved by targeting CD20 or CD22 after the failure of targeting CD19. There were a small number of patients (exact number undisclosed) with CD19-negative/dim expression after treatments with CD19 Si-CAR T-cell therapy who responded to CD22 Si-CAR T-cell therapy [62]. As shown in Table 5, regardless of the small number to date, targeting CD22 or CD20 with Si-CAR T-cell therapy or Bi-CAR T-cell therapy could have helped over twenty patients with CD19 escape achieved CR, among which four patients remained in CR for more than 6 months with 12-month remission in one patient [14,27,28,71]. In particular, seven patients with prior exposure to CD19 Si-CAR T-cell therapy were enrolled in two studies on CD19/CD20 Bi-CAR T-cell therapy [27,28], among whom five patients managed to achieve CR after Bi-CAR T-cell therapy [28]. There were patients with CD19 antigen escape or prior usage of CD19 Si-CAR T-cell therapy who achieved CR after administration of alternative Bi-CAR T cells targeting CD19/CD20 or CD19/CD22. Therefore, clinical data are available to support the targeting of CD10/CD20 or CD19/CD22 in relapsed patients due to resistance to CD19 Si-CAR T-cell therapy.

Overcoming BCMA-negative or BCMA-low escape has been proposed as capable of reversing the resistance of malignant plasma cells to BCMA Si-CAR T-cell therapy [72]. However, after examining more than 200 patients treated with BCMA Si-CAR T-cell therapy in several trials [8,73,74,75,76,77,78,79], BCMA-negative cells were detected in only two patients who relapsed. A BCMA-negative plasma cell population was present in one patient [75], while BCMA-negative and BCMA-positive plasma cells were present in the other patient [80]. By comparison, 10 patients relapsed with BCMA-positive expression or BCMA expression returning to the baseline level [78,81]. Therefore, evidence of relapse resulting from loss or down-regulation of BCMA expression derived from current clinical data is scarce, making the evidence of BCMA-negative or BCMA-low escape not as robust as that for CD19.

Despite limited evidence on the failure of response due to BCMA escape among trials with Si-CAR T-cell therapy targeting BCMA, the combination of BCMA CAR and a second CAR is still being explored in MM [11,12], most of which adopt cocktail/sequential infusion of BCMA Si-CAR T cells and other Si-CAR T cells. Bi-CAR T-cell therapy with bivalent CAR recognizing BCMA/CD19 [82] and BCMA/CD38 [38] have advanced into clinics, while BCMA/CS1(SLAMF7) [42,43] and BCMA/GPRC5D Bi-CAR T-cell therapies [39] are forthcoming, as they were found to be effective in preclinical models. BCMA/CD19 Bi-CAR T-cell therapy showed exciting efficacy in a small group of patients [82]. ORR in five patients was 100%, similar to that reported in most early trials with fewer than 20 patients [74,82]. Only one grade 3 cytokine release syndrome (CRS) occurred without the incidence of neurotoxicity (NT). Data on the DOR and PFS for BCMA/CD19 Bi-CAR T-cell therapy are pending, as data only revealed that the response in one patient with stringent CR (sCR) was >4 months [82]. In addition, BCMA/CD38 Bi-CAR T-cell therapy has also received greater attention due to encouraging clinical results in recent years. After treatments with BCMA/CD38 Bi-CAR T-cell therapy, a responder remained in sCR for >12 months, and five of eight patients with sCR maintained sCR at a median follow-up of 9 months, with the 9-month PFS being 75% [83]. Given that the DOR of present BCMA Si-CAR T-cell therapy in MM is far from satisfactory, Bi-CAR T-cell therapy targeting other antigens together with BCMA might warrant further investigation. Although clinical efficacy, such as response and survival, has been reported to be irrelevant to BCMA expression [73,84], the data on the detailed expression pattern over time in responders who relapsed are limited. Meanwhile, it is unclear whether patients with reduced BCMA expression have been enrolled in current trials of Bi-CAR T-cell therapy. It may be worthwhile to design trials that include the tracking of BCMA expression and the related response in the individual patient during the clinical course, especially among patients treated with Bi-CAR T cells after relapse of BCMA Si-CAR T-cell therapy, to identify patients with reduced BCMA expression compared to baseline who could benefit from BCMA Bi-CAR T-cell therapy.

### 3.2. Expansion of Dual-Targeting CAR T Cells in Hematological Malignancies

In vivo expansion of CAR T cells in patients with hematological malignancies has been summarized in Table 3 and Table 4 to assess if there are any obvious differences in cell proliferation between Si-CAR T cells and dual-targeting CAR T cells. Time to maximum expansion is comparable in ALL and NHL between Si-CAR T cells and dual-targeting CAR T cells, ranging from 9 to 14 days. Similarly, maximum expansion of CAR T cells detected by polymerase chain reaction is comparable in NHL between one CD19 Si-CAR T-cell product [1] and two CD19/CD22 Loop Bi-CAR T-cell products [31,34]. Unfortunately, the disclosed data are insufficient to make a full comparison on the maximum expansion of CAR T cells among different dual-targeting CAR T cells and Si-CAR T cells.

In particular, CD22 CAR T cells in Bi-CAR T cells produced by co-transduction of two vectors expanded poorly [14], which was consistent with the findings from the other groups [33]. When simulating the cocktail of CD19 Si-CAR T cells and CD20 Si-CAR T cells by coculture of two Si-CAR T cells with tumor cells, it was found that CD19 Si-CAR T cells are preferentially amplified over CD20 Si-CAR T cells in vitro [21], which could lessen the effect of CD20 Si-CAR T cells in eliminating CD19-negative cells. However, poor Si-CAR T-cell expansion after the second infusion of Si-CAR T cells was not observed in clinical studies on cocktail/sequential infusion of two Si-CAR T-cell products, one with CD22 Si-CAR T cells after CD19 Si-CAR T cells [15] and the other with mainly CD22 Si-CAR T cells prior to infusion of CD19 Si-CAR T cells [16].

### 3.3. Clinical Safety Profile of Dual-Targeting CAR T-Cell Therapy in the Treatment of Hematological Malignancies

Theoretically, dual-targeting CAR T-cell therapy can be stimulated by two antigens, raising the question of whether stronger activation of CAR T cells in patients than Si-CAR T-cell therapy would occur. Whether dual stimulation in T cells would lead to increased activation of T cells in patients and, therefore, a greater incidence of adverse events than Si-CAR T-cell therapy, requires investigation. Results from early trials enrolling less than 10 subjects to large trials with an enrollment of more than 200 subjects are listed in Table 6 and Table 7. CRS and NT are the main focus in the present review.

Grade 3–4 CRS was absent in the few studies on both Si-CART [62,71,85] and Bi-CART [29,70,86], while no grade 3-4 NT was reported in trials on both Si-CART [3,30,60,62,71] and Bi-CART [28,29,30,86]. For studies with available ASTCT scales for CRS and NT, no grade 3–4 CRS was reported in two of nine (22.2%) trials of Si-CAR T-cell therapy or in three of eight (37.5%) trials of Bi-CAR T-cell therapy. Meanwhile, no grade 3–4 NT was reported in three out of seven trials (42.9%) of Si-CAR T-cell therapy or in four of eight (50%) trials on Bi-CAR T-cell therapy. In conclusion, a higher incidence of grade 3–4 CRS and NT occurred in Si-CAR T-cell therapy than in Bi-CAR T-cell therapy.

The incidence of grade 1–2 CRS was similar between Si-CAR T-cell therapy and Bi-CAR T-cell therapy. All patients experienced a grade 1–2 NT in one trial on CD19 Si-CAR T-cell therapy [65], and all patients experienced grade 1–2 CRS in one trial on CD19/CD22 Bi-CAR T-cell therapy [29], both of which enrolled less than 10 subjects. The incidence of grade 1–2 NT for Si-CAR T-cell therapy was higher than that for Bi-CAR T-cell therapy.

Taken together, these findings indicated that Bi-CAR T-cell therapy is less likely to cause severe CRS and NT than Si-CAR T-cell therapy. There seems to be a difference in the safety profile with respect to the occurrence of CRS and NT between Bi-CAR T-cell therapy and Si-CAR T-cell therapy by simply looking at the numbers; however, considering the sample size, different clinical sites, and possible inadequate management of CRS and NT during early development of CAR T-cell therapy, it seems more investigations are needed to confirm this conclusion.

## 4. Comparison of Different Dual-Targeting CAR T-Cell Therapies

As shown in Figure 2, different dual CAR strategies have been translated into six clinical trials on Tandem Bi-CAR T cells targeting CD19/CD20 [27,28], CD19/CD22 [29,30,64], and BCMA/CD38 [38]; four trials on cocktail/sequential infusion of two separate Si-CAR T cells of targeting CD19 or CD22 [15,16] and BCMA or CD19 [88,89]; three trials on Loop Bi-CAR T cells targeting CD19/CD22 [31,34,90]; two trials on bicistronic Bi-CAR T cells targeting CD19/CD22 [32,70]; and two trials on CD19/CD22 Bi-CAR T cells produced by co-transduction of two separate vectors [14,91]. Because the exact construct (tandem or loop) for DOR and survival of a dual BCMA/CD19 targeted FasT CAR-T GC012F [92] has not been disclosed so far, the trial was not included in this section.

No matter how effective dual CAR strategies have been in preclinical models, it may only be considered a true success when it benefits patients in the clinic. Emerging clinical data in 2021 permit a fair comparison of different dual-targeting CAR T-cell therapies possible in patients (Table 8).

When comparing the safety profiles in trials with a sample size of 10 patients, grade 3–4 CRS occurred in 8% to 13.3% of patients given Tandem Bi-CAR T cells [30,64,87], 11% in patients given Bi-CAR T cells produced by co-transduction of two vectors [14], and in none of the patients given bicistronic Bi-CAR T cells [70]. Grade 3–4 NT occurred in 2% to 8% of patients given Tandem Bi-CAR T cells [64,87], 4% in patients given Bi-CAR T cells produced by co-transduction of two vectors [14], and 5.7% in patients given bicistronic Bi-CAR T cells [70]. There was no difference in terms of safety profiles among different dual-targeting CAR T-cell therapies. Hence, it is important to compare the ultimate criteria, DOR, and survival among patients treated with different dual-targeting CAR T-cell therapies.

During the optimization of the CD19/CD22 Bi-CAR T construct, the loop construct was determined to be superior to the tandem one and, thus, moved forward to the clinical phase [33,34]. When comparing the clinical outcomes in ALL, the PFS in the trial using Loop Bi-CAR T cells [34] was longer than the LFS in the trial using Tandem Bi-CAR T cells [30], whereas OS in the former was shorter than that in the latter. It is noteworthy that the spatial structure of CD19 scFv and CD22 scFv in the Bi-CAR was different, although the transduction efficiencies were comparable between studies. Moreover, the number of patients who proceeded to HSCT after Bi-CAR T-cell therapy differed between studies. Similar outcomes were found between Bicistronic Bi-CAR T-cell threapy and Loop Bi-CAR T-cell therapy. Although the transduction efficiencies of the CD19/CD22 Bicistronic Bi-CAR T-cell product [32] were much lower than those of CD19/CD22 Loop Bi-CAR T-cell product [34] in patients with ALL, survival was better in the former. So far, the Loop Bi-CAR T-cell product has not shown superiority over Tandem Bi-CAR T-cell product and Bicistronic Bi-CAR T-cell product to benefit patients with ALL. The comparisons should be considered with caution because they are not derived from head-to-head studies.

When comparing clinical outcomes on CD19/CD22 Loop Bi-CAR T-cell therapy with different locations of CD19 scFv and CD22 scFv on CAR in NHL subjects, the PFS in the trial on CAR T cells expressing Bi-CAR with CD19 scFv distal to 4-1BB [31] was longer than the one with CD22 scFv distal to 4-1BB [34], both of which have similar OS. Of note, the transduction efficiency of the CD19/CD22 Loop CAR with CD19 scFv distal to the 4-1BB was lower than the lowest one of the CD19/CD22 Loop CAR with CD22 scFv distal to 4-1BB, indicating that optimization may still be needed. Transduction efficiencies of CD19/CD20 Tandem Bi-CAR T-cell product in patients with NHL dropped to nearly one-third of the those in vitro [28]. However, the PFS and OS rates of patients given CD19/CD20 Tandem Bi-CAR T-cell therapy were higher than those with CD19/CD22 Loop Bi-CAR T-cell therapy, despite higher transduction efficiencies observed in the latter [28,34]. Of course, caution is needed to interpret non-head-to-head studies, and the differences may be due to different targets in NHL. Overall, poor transduction efficiencies may not necessarily worsen clinical outcomes, though improving transduction efficiencies still matters in optimization.

Despite a great deal of effort directed at optimizing the Bi-CAR T-cell product, the outcomes have not been able to outperform the simple strategy of the cocktail/sequential infusion of two Si-CAR T-cell products without relentless optimization. For ALL and NHL, the cocktail/sequential infusion of CD19/CD22 Si-CAR T-cell products achieved the longest median OS [16], not only providing convincing clinical evidence of dual-targeting CAR T-cell therapy to improve survival but also dwarfing other time- and cost-consuming trials from preclinic to clinic. It is time for different research groups to collaborate and share details on optimizing Bi-CAR structure and standardize clinical trials to compare different dual-targeting therapeutic strategies in a quest for the ideal construct to produce Bi-CAR. Meanwhile, the cocktail/sequential infusion of two Si-CAR T-cell products still merits clinical application to save the lives of patients with ALL and NHL if the commercialization of other dual CAR strategies requires additional time.

In terms of experience gained from the cocktail/sequential infusion of two Si-CAR T-cell products, the timing of the second infusion warrants further exploration. During the trial on the cocktail infusion of BCMA/CD19 Si-CAR T-cell products, CD19 Si-CAR T and BCMA Si-CAR T-cell product were infused on the same day [89]. During the cocktail/sequential infusion of CD19/CD22 Si-CAR T-cell products, CD22 Si-CAR T cells were infused one day before CD19 Si-CAR T cells [16]. Comparing the cocktail infusion of BCMA/CD19 Si-CAR T-cell products [89] with BCMA/CD38 Tandem Bi-CAR T-cell product [38], the PFS of the former was much shorter than that of the latter. Again, the variance may be attributed to the difference in targets. However, it may be worthwhile to adjust the timing of the cocktail/sequential infusion of two Si-CAR T-cell products to standardize the comparison and investigate the influence of the timing of the infusion on the expansion of two different Si-CAR T-cell products in preclinical models.

## 5. Challenges and Perspectives

Advancing technologies have made Bi-CAR T-cell therapy readily available; however, three main limitations remain for Bi-CAR T-cell therapy: (1) Bi-CAR T-cell therapy does not address other proposed resistance mechanisms outside of target antigen loss; (2) evidence on the safety profile and in vivo activity of Bi-CAR T cells are insufficient; and (3) increased difficulty in manufacturing since the size of construct is bigger. The specific challenges within Bi-CAR T cell manufacturing are the complicated optimization process to find the suitable vectors for manufacturing, increased inconsistency in batch manufacture of viral vector, low transduction efficiency in Bi-CAR T cells, and high manufacturing failure rate due to the size of the bivalent and bicistronic vector.

The limitations of the review are as follows: (1) For earlier phase studies, PFS/OS estimations are based on small numbers of patients. It should also be taken into account that patients may receive additional treatment after CAR T-cell therapy. (2) Comparison of efficacies of different approaches must take into account wide confidence intervals of PFS/OS estimation. (3) Comparison of safety profiles must take into account differences in scoring systems for toxicity and changes in toxicity management over time.

In conclusion, dual-targeting CAR T-cell therapy has offered another hope for patients in the post era after the use of Si-CAR T-cell therapy. The clinical efficacy has been validated in trials on cocktail/sequential infusion of Si-CAR T-cell products and in a few trials of Bi-CAR T-cell therapy. However, an optimal Bi-CAR structure has not been established. The pooled safety profiles of Bi-CAR T-cell appear better than those of Si-CAR T-cell therapy, with a lower incidence of severe CRS and NT. No apparent effect of 1 + 1 > 2 in terms of DOR, OS, and PFS has been demonstrated in trials on Bi-CAR T-cell therapy, indicating that further optimization is needed. The optimization of the Bi-CAR should focus on finding the right targets for different indications, the appropriate spatial structure of two different scFvs, the suitable linker for scFvs, and the proper transduction efficiencies using patients’ T cells to enhance the efficacy and the persistence of Bi-CAR T cells in patients.

The lack of a magic bullet as Bi-CAR structure calls for collaboration of different research groups to develop solutions to benefit the global community. Models integrating clinical data with preclinical data to predict the optimal Bi-CAR may help design an ideal vector for Bi-CAR introduction. Meanwhile, more trials with Bi-CAR T-cell therapy in patients without prior exposure to Si-CAR T-cell therapy are also needed to compare the two types of CAR T-cell therapy.

## Figures and Tables

**Figure 1 cancers-14-03230-f001:**
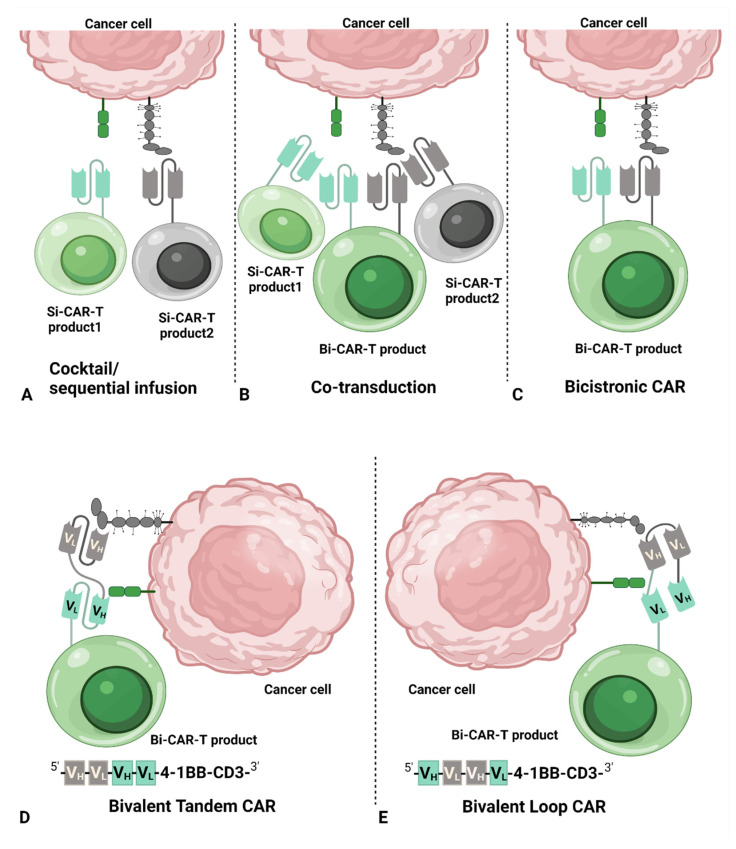
Illustration of dual-targeting CAR T-cell strategies. Dual-targeting CAR T-cell therapy is a therapeutic strategy to identify two tumor-associated antigens on cancer cells, which can be categorized into five dual CAR strategies. (**A**) Cocktail/sequential infusion of two Si-CAR T-cell products individually transduced with two different vectors. (**B**) A pool of two Si-CAR T-cell products and one Bi-CAR T-cell product by co-transductions of two vectors each encoding one individual CAR. (**C**) one Bi-CAR T-cell product produced by transduction of a bicistronic vector to introduce two separate CARs with one antigen-binding domain per CAR. (**D**,**E**) One Bi-CAR T-cell product expressing one bivalent CAR with two antigen-binding domains. Bivalent CAR can be categorized into two different structures by placing the V_L_ and V_H_ of scFv in different order, i.e., with V_L_-V_H_ of one scFv directly linked to the V_L_-V_H_ of the other scFv defined as bivalent tandem CAR (**D**) or with V_L_-V_H_ of one scFv separated by the V_L_-V_H_ of the other scFv defined as bivalent loop CAR (**E**). Illustration was created with BioRender.com. Abbreviations: Bi-CAR-T, bispecific chimeric antigen receptor T-cell; CAR, chimeric antigen receptor; scFv, single-chain variable fragments; Si-CAR-T, single-targeted chimeric antigen receptor T-cell; V_H_, variable heavy chain; V_L_, variable light chain.

**Figure 2 cancers-14-03230-f002:**
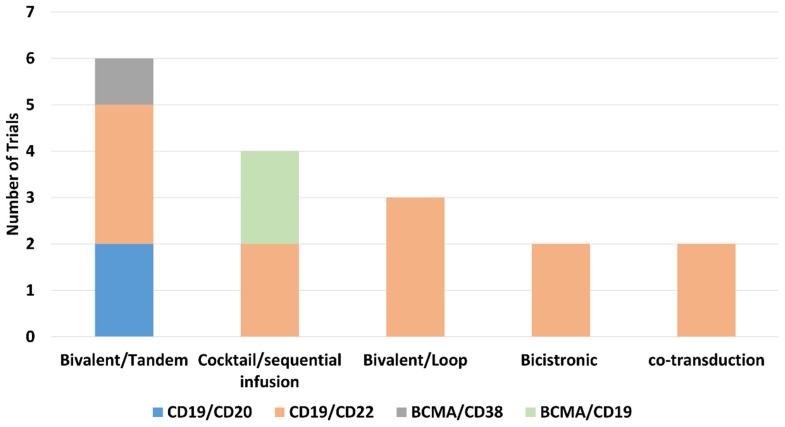
Clinical trials of different dual-targeting CAR T-cell therapies. Different dual CAR strategies have been translated into six clinical trials on Tandem Bi-CART targeting CD19/CD20, CD19/CD22, and BCMA/CD38; four trials on cocktail/sequential infusion of two separate Si-CAR-T products on CD19/CD22 Si-CART and BCMA/CD19 Si-CART; three trials on Loop Bi-CART targeting CD19/CD22; two trials on bicistronic Bi-CART targeting CD19/CD22; and two trials on CD19/CD22 Bi-CART produced by co-transduction of two separate vectors. The bar chart was created using Microsoft^®^ Excel^®^ version 2111. Abbreviations: BCMA, B cell maturation antigen; Bi-CART, bispecific chimeric antigen receptor T cells; Bi-CAR-T, bispecific chimeric antigen receptor T-cell; CAR, chimeric antigen receptor; CAR-T, chimeric antigen receptor T cell(s); Si-CART, single-targeted chimeric antigen receptor T cells; Si-CAR-T, single-targeted chimeric antigen receptor T-cell.

**Table 1 cancers-14-03230-t001:** Advantages and disadvantages of common dual CAR strategies.

CAR Strategy	Advantages	Disadvantages
Cocktail/ sequential infusion of two Si-CAR T-cell products	Availability of optimized single CAR construct, vectors, and transduction process Precise dose for each Si-CAR T-cell product	High manufacturing cost due to doubling the cost of producing and quality control. Uneven expansion Optimization of the timing of second infusion
Co-transduction with two Si-CAR vectors	Availability of optimized CAR construct and vectors	Optimization of the transduction process High manufacture cost due to twice the amount of vectors and viruses required Heterogeneity of cell products mixed with Si-CAR T cells and Bi-CAR T cells Uneven expansion of Si-CAR T cells
Bicistronic Bi-CART	Dual co-stimulation Homogeneity of cell products	Large vector size Low transduction efficiency
Bivalent Tandem Bi-CART	Reduced manufacture cost Homogeneity of cell products	Complex construct optimization
Bivalent Loop Bi-CART	Reduced manufacture cost Homogeneity of cell products Higher potency than Tandem	Complex construct optimization

Abbreviations: Bi-CART, bispecific chimeric antigen receptor T cells; Bi-CAR-T, bispecific chimeric antigen receptor T-cell; CAR, chimeric antigen receptor; CAR-T, chimeric antigen receptor T cell(s); Si-CAR, single-targeted chimeric antigen receptor; Si-CART, single-targeted chimeric antigen receptor T cells.

**Table 2 cancers-14-03230-t002:** Comparison of transduction efficiencies and effects among different dual CAR strategies in vitro and in vivo.

Ref.: First Author	Target	Stage	Construct/CAR Strategy	Transduction Efficiency	IL-2	IFN-ɣ	Cytotoxicity	In Vivo
Zah [21]	CD19/ CD20	Preclinic	Tandem19-20 long (CD19-LinkerG4S-CD20-HingeCH2CH3-CD28tm-4-1BB-CD3-T2A-EGFRt; HingeCH2CH3=229 aa)	NA	∼0 (CD19 K562); ∼200 pg/Ml (CD20 K562)	∼1000 pg/Ml (CD19 K562); ∼2200 pg/Ml (CD20 K562)	∼11% (E:T = 10:1, CD20 K562)	(Only comparing Si-CART with Bi-CART)
			Tandem20-19 long (CD20-LinkerG4S-CD19-HingeCH2CH3-CD28tm-4-1BB-CD3-T2A-EGFRt; HingeCH2CH3=229 aa)		∼0 (CD19 K562); ∼10 pg/mL (CD20 K562)	∼1800 pg/mL (CD19 K562); ∼2000 pg/mL (CD20 K562)	∼13% (E:T = 10:1, CD20 K562)	
			Tandem19-20 short (CD19-LinkerG4S-CD20-Hinge-CD28tm-4-1BB-CD3-T2A-EGFRt; Hinge=12 aa)		∼1400 pg/mL (CD19 K562); ∼200 pg/mL (CD20 K562)	∼3800 pg/mL (CD19 K562); ∼600 pg/mL (CD20 K562)	∼21% (E:T = 10:1, CD20 K562)	
			Tandem20-19 short (CD20-LinkerG4S-CD19-Hinge-CD28tm-4-1BB-CD3-T2A-EGFRt; Hinge=12 aa)		∼1500 pg/mL (CD19 K562); ∼100 pg/mL (CD20 K562)	∼4200 pg/mL (CD19 K562); ∼2100 pg/mL (CD20 K562)	∼35% (E:T = 10:1, CD20 K562)	
			Tandem20-19 short=ii (CD20-LinkerG4Sx4-CD19-Hinge-CD28tm-41BB-CD3-T2A-EGFRt; Hinge=12 aa) *		∼150 pg/mL (CD19- Raji); highest in CD19- Raji	∼2200 pg/mL (CD19- Raji)	∼60% (E:T = 10:1, CD19- Raji) #	
Schneider [26]	CD19/CD20	Preclinic	Tandem1920 (CD19-LinkerGS-CD20-CD8tm-41BB-CD3)	85%	∼2000 pg/mL	∼4000 pg/mL	Tandem2019 > Tandem1920 in various cell lines	Tumor burden 25 days after inoculation: No difference between 2019, 1920 and 19 + 20 co-administration; Survival 25 days after inoculation: 2019 > 19 + 20 co-administration
			Tandem2019 (CD20-LinkerGS-CD19-CD8tm-41BB-CD3) *	89%	∼2200 pg/mL	∼4500 pg/mL		
Shah [27]	CD19/ CD20	Clinic	Tandem (CD20-CD19-CD8 hinge-4-1BB-CD3)	7.4–28%	NA	NA	NA	NA
Tong [28]	CD19/ CD20	Preclinic & Clinic	TanCAR1 (CD19V_L_-CD19V_H_-LinkerEA-CD20V_H_-CD20V_L_-CD8-4-1BB)	22%	<3500 pg/mL	∼1500 pg/mL	<40%	(Only comparing Si-CART with Bi-CART)
			TanCAR2 (CD19V_L_-CD19V_H_-LinkerG4S-CD20V_H_-CD20V_L_-CD8-41BB)	19%	∼4000 pg/mL	∼1600 pg/mL	<60%	
			TanCAR3 (CD19V_H_-CD19V_L_-LinkerEA-CD20V_L_-CD20V_H_-CD8-41BB)	33%	<3500 pg/mL	∼1600 pg/mL	<60%	
			TanCAR4 (CD19V_H_-CD19V_L_-LinkerG4S-CD20V_L_-CD20V_H_-CD8-41BB)	39%	<3500 pg/mL	∼1600 pg/mL	<60%	
			TanCAR5 (CD20V_L_-CD20V_H_-LinkerEA-CD19VH-CD19VL-CD8-41BB)	17%	<3500 pg/mL	∼1500 pg/mL	<60%	
			TanCAR6 (CD20V_L_-CD20V_H_-LinkerG4S-CD19V_H_-CD19V_L_-CD8-41BB)	33%	<3500 pg/mL	∼1500 pg/mL	<60%	
			TanCAR7 (CD20V_H_-CD20V_L_-LinkerEA-CD19V_L_-CD19V_H_-CD8-41BB) *	35% (10.1–35.1% in patients’ PBMC)	∼3500 pg/mL	∼1600 pg/mL	>60% (Raji)	
			TanCAR8 (CD20V_H_-CD20V_L_-LinkerG4S-CD19V_L_-CD19V_H_-CD8-41BB)	33%	<3500 pg/mL	∼1600 pg/mL	<60%	
Dai [29]	CD19/ CD22	Clinic	TanCAR (CD22m971-LinkerEAAAK-CD19FMC63-CD8-4-1BB-CD3)	10.32–16.91%	1700 pg/mL	4000 pg/mL		NA
Wang [30]	CD19/ CD22	Clinic	TanCAR (CD19V_L_-CD19V_H_-CD22V_L_- CD22V_H_-4-1BB-CD3)	60.1 (30–75.1)%	NA	NA	NA	NA
Zhang [31]	CD19/ CD22	Clinic	Loop (CD22V_L_-CD19V_L_-CD19V_H_-CD22V_H_-4-1BB-CD3)	20 to ∼78%	NA	NA	NA	NA
Cordoba [32]	CD19/ CD22	Clinic	Bicistronic	17.7% (8.6–39.3%)	NA	NA	∼100%	Tumor burden in CD19- mice: Bi-CAR-T < CD19 Si-CART
Qin [33]	CD19/ CD22	Preclinic	Co-transduction with two Si-CAR vectors	23%	NA	NA	NA	Tumor burden 13 days after inoculation: TanCAR1 < TanCAR4; (For LoopCAR, only comparing Si-CART with Bi-CART)
			TanCAR1 (CD22V_H_-Linker1G4Sx1-CD22V_L_-L5G4Sx5-CD19V_L_-Linker6TKPE-CD19V_H_-CD8-4-1BB)	60%	∼11,000 pg/mL (CD19CD22 K562)			
			TanCAR2 (CD19V_L_-Linker6TKPE-CD19V_H_-Linker5G4Sx5-CD22V_H_-Linker1G4Sx1-CD22V_L_-CD8-4-1BB)	29%	NA			
			TanCAR3 (CD22V_H_-Linker6TKPE-CD22V_L_-Linker5G4Sx5-CD19V_L_-Linker6TKPE-CD19V_H_-CD8-4-1BB)	0%	NA			
			TanCAR4 (CD22V_H_-Linker1G4Sx1-CD22V_L_-Linker4G4Sx4-CD19V_L_-Linker6TKPE-CD19V_H_-CD8-4-1BB)	56%	∼26,000 pg/mL (CD19CD22 K562)			
			LoopCAR1 (CD19V_L_-Linker3G4Sx3-CD22V_H_-Linker1G4Sx1-CD22V_L_-Linker3G4Sx3-CD19V_H_-CD8-4-1BB)	19%	∼<2000 pg/mL (CD19CD22 K562)			
			LoopCAR2 (CD19V_L_-Linker3G4Sx3A-CD22V_H_-Linker6TKPE-CD22V_L_-Linker3G4Sx3B-CD19V_H_-CD8-4-1BB)	42%	∼2800 pg/mL (CD19CD22 K562)			
			LoopCAR3 (CD19V_L_-Linker2G4Sx2-CD22V_H_-Linker6TKPE-CD22V_L_-Linker2G4Sx2-CD19V_H_-CD8-491BB)	24%	∼25000 pg/mL (CD19CD22 K562)			
			LoopCAR4 (CD22V_H_-Linker2G4Sx2-CD19V_L_-Linker2G4Sx2-CD19V_H_-Linker2G4Sx2-CD22V_L_-CD8-4-1BB)	63%	∼5000–26,000 pg/mL (CD19CD22 K562)			
			LoopCAR5 (CD19V_L_-Linker3G4Sx3C-CD22V_H_-Linker2G4Sx2-CD22V_L_-Linker3G4Sx3D-CD19V_H_-CD8-4-1BB)	49%	∼10,000 pg/mL (CD19CD22 K562)			
			LoopCAR6 (CD19V_L_-Linker1G4Sx1-CD22V_H_-Linker6TKPE-CD22V_L_-Linker1G4Sx1-CD19V_H_-CD8-4-1BB) *	82%	∼22,000 pg/mL (CD19CD22 K562)			
Spiegel [34]	CD19/ CD22	Clinic	Loop (CD19V_H_-CD22V_L_-CD22V_H_-CD19V_L_-CD8-4-1BB) *	60.1%	NA	NA	NA	NA
Yang [35]	CD19/ CD22	Preclinic & Clinic	Loop GC022C	67.50%	NA	NA	75% (1:1)	NA
			Loop GC022F	53.60%	NA	NA	55% (1:1)	NA
Wang [16]	CD19/ CD22	Clinic	Cocktail/Sequential infusion of two Si-CAR-T products with separate Si-CAR vectors	40.4% ± 18.4% (CAR19); 42.8% ± 19.6% (CAR22)	∼3500 pg/mL (Raji)	∼15,000 pg/mL(Raji)	∼60% CD22;∼50% CD19 (E:T = 10:1; Raji)	Reducing Leukemia burden: infusion of one Si-CAR-T product ∼ co-infusion of two Si-CAR-T products
Pan [15]	CD19/ CD22	Clinic	Sequential infusion of two Si-CAR-T products with separate Si-CAR vectors	10.4%∼74.7% (CAR19); 8.3%∼69.8% (CAR22)	NA	NA	NA	NA
Ruella [36]	CD19/ CD123	Preclinic	Bicistronic	46%	NA	NA	NA	NA
Kang [37]	BCMA/ CD19	Preclinic	Tandem (BCMA-CD19-CD8tm-CD28-CD3)	46% to 55%	NA	NA	NA	NA
Mei [38]	BCMA/ CD38	Preclinic	Tandem 38BM (CD38-BCMA-CD8-4-1BB-CD3)	60.1%	NA	BM38 > 38BM	BM38 > 38BM	Survival: BM38 > 38BM
			Tandem BM38 (BCMA-CD38-CD8-4-1BB-CD3)	59.4%				
		Clinic	Tandem BM38 (BCMA-CD38-CD8-4-1BB-CD3)	12% to 60%	NA	NA	NA	NA
de Larrea [39]	BCMA/ GPRC5D	Preclinic	Co-infusion of two Si-CAR-T products with separate Si-CAR vectors	60% to 70%	NA	NA		Efficacy: Bicistronic = separate Si-CAR vectors > Tandem in BCMA-GPRC5D+ models; Tandem > Bicistronic > separate Si-CAR vectors in BCMA+ GPRC5D+ models
			Bicistronic (BCMA-4-1BB-GPRC5D-41BB)	60% to 70%	NA	NA	∼80% (BCMA-/GPRC5D+)	
			Bicistronic (BCMA-4-1BB-GPRC5D-CD28)	60% to 70%	NA	NA	∼65% (BCMA-/GPRC5D+)	
			Tandem (GPRC5D-BCMA-4-1BB)	60% to 70%	NA	NA	∼65% (BCMA-/GPRC5D+)	
Globerson [40]	CD138/ CD38	Preclinic	Bicistronic (CD138V_L_-Linker-CD138V_H_-CD28-CD38V_L_-CD38V_H_-CD8-FcγR)	72%		2000–3000 pg/mL	∼90%(E:T = 1:1)	97.4 days (*n* = 26)
Dai [41]	CD5/ CD7	Preclinic	bicistronic (CD7-4-1BB-CD3-P2A-CD5-4-1BB-CD3-T2A-EGFRt)	12.4%, 34.2%	NA	NA in concentrations	Tan5-7 =Tan7-5 > bicistronic	Expansion and persistence: Tan5-7 = Tan7-5 > bicistronic
			Tan5-7 (CD5-Linker-CD7-4-1BB-CD3-T2A-EGFRt)	58.1%, 62.2%	NA	NA in concentrations		
			Tan7-5 (CD7-Linker-CD5-4-1BB-CD3-T2A-EGFRt)	49%, 57.6%	NA	NA in concentrations		
Zah [42]	BCMA/ CS1 (SLAMF7)	Preclinic	TanCS1-BCMA (CS1-LinkerG4S-BCMA-Hinge-CD28tm-41BB-CD3-T2A-EGFRt, 1122aa)	∼41%	NA	NA	Si-CART < Bi-CART	Survival: TanCS1-BCMA = TanBCMA-CS1
			TanBCMA-CS1 (BCMA-LinkerG4S-CS1-Hinge-CD28tm-41BB-CD3-T2A-EGFRt, 1121aa)	∼35%	NA	NA		
			bicistronic (CS1-BCMA, 1194aa and 1411aa)	0.97% to 2.56%	NA	NA		
Chen [43]		Preclinic	bicistronic (BCMA-CS1)	19.89%	NA	NA	NA	NA

*: Bi-CART that is considered the optimal one. # In CD19- Raji cells, Tandem20-19 short (CD20-LinkerG4S-CD19-Hinge-CD28tm-41BB-CD3-T2A-EGFRt; Hinge=12 aa) was approximately 40%, while Tandem20-19 short=ii (CD20-LinkerG4Sx4-CD19-Hinge-CD28tm-41BB-CD3-T2A-EGFRt; Hinge=12 aa) was approximately 60%. Abbreviations: aa, amino acid; Ref., reference; NA, not available; V_H_, variable heavy chain; V_L_, variable light chain.

**Table 5 cancers-14-03230-t005:** Outcomes in patients with negative or low-CD19 expression after treatments with CD22 Si-CAR T-cell therapy and CD19/CD20 Bi-CAR T-cell therapy.

Ref.: First Author	Target	Characteristics of CD19 and CD22 Expression	Outcome
Fry [71]	CD22	10 ALL pts with CD19neg or CD19dim	CR: 6/10 *, 4 in CR for ≥ 6 mon; 1 in CR for 12 mon; 1 in CR for 9 mon ongoing
Tong [28]	CD19/CD20	4 NHL pts with CD19neg	CR: 2/4; PR: 1/4; PD:1/4
Shah [27]	CD19/CD20	4 NHL pts with < 40% CD19	CR: 3/4; PR: 1/4
Gardner [14]	CD19/CD22	13 ALL pts with diverse expression of CD19 and CD22	CR: approximately 9–11/13

*: amount of CR/number of CD19neg pts. Abbreviations: ALL, acute lymphoblastic leukemia; CR, complete response; mon, month(s); NA, not available; NR, not reached; neg, negative; NHL, non-Hodgkin lymphoma; OS, overall survival; PR, partial response; PD, progressive disease; Ref., reference; pts, patients.

**Table 6 cancers-14-03230-t006:** Comparing CRS and NT in dual-targeting CAR T-cell therapy with Si-CAR-T therapy in ALL.

Ref.: First Author	Target	Enrollment	CRS Gr1-2	CRS Gr3-4	NT Gr1-2	NT Gr3-4
Maude [60]	CD19	30	22/30 (73%)	8/30 (27%)	13/30 (43%)	None
Maude [3]	CD19	75	77%	∼25%	30/75 (40%)	None
Wang [30]	CD19	35	19/35 (54.3%)	16/35 (45.7%)	2/35 (5.7%)	None
Fry [71]	CD22	21	16/21 (76%)	None	Mild/transient/mild-moderate >2/21 (10%)	
Shah [62]	CD22	58 *	45/58 (90%)	12/58 (24%)	minimal/transient	
Dai [29]	CD19/CD22	6	100%	None	None	None
Schultz [64]	CD19/CD22	12	9/12 (75%)	1/12 (8%)	2/12 (17%)	1/12 (8%)
Wang [30]	CD19/CD22	15	13/15 (86.7%)	2/15 (13.3%)	None	None
Wang [16]	CD19/CD22	51	40/51 (78.4%)	11/51 (21.6%) ∫	11/51 (12%)	1/51 (1%)
Pan [15]	CD19/CD22	20	17/20 (85%)	1/20 (5%)	3/20 (15%)	1/20 (5%)
Spiegel [34]	CD19/CD22	17	12/17 (70.6%)	1/17 (5.9%)	2/17 (11.8%)	3/17 (17.6%)
Cordoba [32]	CD19/CD22	15	12/15 (80%)	0	4/15 (26.7%)	0

* 56 ALL, 1 diffuse large B-cell lymphoma, 1 chronic myeloid leukemia. ∫ denotes Gr 3-5. Abbreviations: ALL, acute lymphoblastic leukemia; CRS, cytokine release syndrome; Gr, grade; NT, neurotoxicity.

**Table 7 cancers-14-03230-t007:** Comparison of CRS and NT in dual-targeting CAR T-cell therapy with Si-CAR-T therapy in NHL.

Ref.: First Author	Target	Enrollment	CRS Gr1-2	CRS Gr3-4	NT Gr1-2	NT Gr3-4
Locke [65]	CD19	7	5/7 (71%)	1/7 (14%)	100%	4/7 (57%)
Jacobson [68]	CD19	148	111/148 (75%)	10/148 (7%)	59/148 (40%)	28/148 (19%)
Abramson [85]	CD19	28	10/28 (36%)	None	5/28 (18%)	4/28 (14%)
Abramson [1]	CD19	269	∼40%	6/269 (2%)	∼30%	27/269 (10%)
Zhang [69]	CD20	11	None severe			
Shah [86]	CD19/CD20	11	6/11 (55%)	None	3/11 (27%)	None
Shah [27]	CD19/CD20	22	14/22 (64%)	1 (5%)	7/22 (32%)	3 (14%)
Tong [28]	CD19/CD20	28	∼30%	4/28 (14%)	∼14%	None
Zhang [87]	CD19/CD20	87	61%	10%	NA	2%
Tholouli [70]	CD19/CD22	35	12/35 (34%)	None	1/35 (3%)	2/35 (5.7%)
Wang [16]	CD19/CD22	38	30 (78.9%)	8 (21.1%)∫	NA	NA
Zhang [31]	CD19/CD22	32	20 (62.5%)	9 (28.1%)	1 (3.1%)	4 (12.5%)
Spiegel [34]	CD19/CD22	21	15/21 (71.4%)	1/21 (4.8%)	8/21 (38.1%)	1/21 (4.8%)

∫ denotes Gr 3-5. Abbreviations: CRS cytokine release syndrome; Gr grade; NT neurotoxicity; NHL, non-Hodgkin lymphoma.

**Table 8 cancers-14-03230-t008:** Comparison of optimization process, transduction efficiencies, DOR, and OS among different dual-targeting CAR T-cell therapies (*n* > 10; ALL and NHL).

Ref.: First Author	Target	CAR Strategy	Optimization Process	Final CAR Transduction Efficiency (Normal Donor vs. Patient)	Durability	OS (mon and %)
Schneider [26], Shah [27]	CD19/CD20	One Tandem Bi-CAR-T product	2 constructs Change order of CAR19 and CAR20 Final: CD20 scFv distal to 4-1BB	85%–89% vs. 7.4–28%	NHL: 12 CR > 6 mon; 6 CR > 12 mon; 8 CR ongoing	NHL: NA
Tong [28]	CD19/CD20	One Tandem Bi-CAR-T product	8 constructs Change order of CAR19 and CAR20 Final: CD20 scFv distal to 4-1BB	35% vs. 10.1%–35.1%	NHL: 64% (12-mon PFS)	NHL: 71% (12-mon OS)
Wang [30]	CD19/CD22	One Tandem Bi-CAR-T product	Undisclosed Final: CD19 scFv distal to 4-1BB	Undisclosed vs. 60.1 (30–75.1)%	ALL: ∼3 mon (median LFS in 13 non-HSCT pts)	ALL: ∼21 mon (median OS in all pts)
Wang [16]	CD19/CD22	Cocktail/Sequential infusion of two Si-CAR products	Not required	52.2% vs. 40.4% ± 18.4% (CAR19); 53.8% vs. 42.8% ± 19.6% (CAR22)	ALL: 52.9% (12-mon PFS) 13.6 mon (median PFS) NHL: 9.9 mon (median PFS) 50.0% (12-mon PFS)	ALL: 62.8% (12-mon OS) 31 mon (median OS) NHL: 18.0 mon (median OS) 55.3% (12-mon OS)
Qin [33], Spiegel [34]	CD19/CD22	One Loop Bi-CAR-T product	Co-transduction vs. 4 Bivalent/Tan constructs vs. 6 Loop constructs Final: CD22 scFv distal to 4-1BB	82% vs. 60.1% (34.6–75.2%)	ALL: 5.8 mon (PFS) ∼0% (12-mon PFS) NHL: 3.2 mon (PFS) ∼25% (12-mon PFS)	ALL: 11.8 mon (median OS in all pts) ∼25% (12-mon OS) NHL: 22.5 mon (median OS) ∼64% (12-mon OS)
Zhang [31]	CD19/CD22	One Loop Bi-CAR-T product	Undisclosed Final: CD19 scFv distal to 4-1BB	Undisclosed vs. 20-(∼)78%	NHL: 40.0% (12-mon PFS) 66.7% (12-mon PFS in CR at 3 mon)	NHL: 63.3% (12-mon OS) 100% (12-mon OS in CR at 3 mon)
Cordoba [32]	CD19/CD22	One Bicistronic Bi-CAR-T product	Binder humanization	56.8% vs. 17.7% (8.6–39.3%)	ALL: 32% (12-mon EFS)	ALL: 60% (12-mon OS)

Abbreviations: ALL, acute lymphoblastic leukemia; CR, complete response; DOR, duration of response; EFS, event-free survival; f/u, follow-up; LFS, leukemia-free survival; mon, month(s); NA, not available; non-HSCT, no hematopoietic stem cell transplantation; OS, overall survival; PFS, progression-free survival; pts, patients; Ref., reference; RFS, relapse-free survival.

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
