# Peer review of "Current Status and Perspectives of Dual-Targeting Chimeric Antigen Receptor T-Cell Therapy for the Treatment of Hematological Malignancies"

_cancers, 2022, doi:10.3390/cancers14133230_

Round 1

Reviewer 1 Report

Bailu Xie and colleagues summarize our knowledge on dual-targeting CAR T cells for the treatment of hematological malignancies. Due to the high number of patients relapsing with antigen-low or antigen-negative disease following CAR T cell therapy, dual-targeting CAR T cells are a promising tool to improve the durability of the anti-tumor response and thus survival of patients. The authors comprehensively discuss different categories of dual CAR T cells, clinical response rates and safety of dual CAR T cells in comparison of monospecific CAR T cells and conclude with challenges that have to be overcome to successfully translate dual CAR T cells into the clinics. Before publication of the review article following changes/suggestions should be considered:  

-       Line 77: The main method for the generation of CAR T cells is lentiviral or retroviral transduction. The authors should be more consistent with the terms transfection and transduction (transfection is the introduction of genetic material using a non-viral approach and therefore not correct in this context)

-       Line 84-86: Rephrase the product of co-transduction to enhance comprehension

-       Line 96: Do all described tandem/loop CARs incorporate 41BB? Or are there also tandem/loop CARs with CD28 or other costimulatory domains?

-       Line 102: Introduce bicistronic CAR before tandem and loop CAR structure because it’s more commonly used and better matches the figure 

-       Line 139: Transduction NOT transfection

-       Line 145: Product without s and delete products in line 147. The sentence should be “A pool of two Si-CAR-T products and one Bi-CAR-T product…”

-       It should be discussed that tandem/loop CARs have one intracellular domain whereas bicistronic CARs have two intracellular T cell activating domains. Would that result in better activation of CARs when both antigens are presented? Are there any data investigating this hypothesis?

-       Figure 2: Are these clinical trials registered at clinicaltrial.gov? 

-       Figure 2: Should be included later in the manuscript when the clinical trials are discussed

-       Line 174-176: Please add short explanations for OR-gate, AND, NOT and synNotch strategy

-       Line 191: Transduction not transfection

-       Line 265: The authors should consider adding trogocytosis as a mechanism of CAR T failure

-       Line 276-78: Please rephrase this sentence 

-       Line 294-296: Are these ALL patients that have been treated with CART19? 

-       Line 341-343: Please rephrase this sentence to improve comprehension

-       Line 378-379: Please rephrase this sentence to improve comprehension

Author Response

Response to Reviewer 1 Comments

Point 1-       Line 77: The main method for the generation of CAR T cells is lentiviral or retroviral transduction. The authors should be more consistent with the terms transfection and transduction (transfection is the introduction of genetic material using a non-viral approach and therefore not correct in this context)

Response 1: Thank you for pointing out the error. “Transfection” and “transfect” have been replaced with “transduction” or “transduce”, respectively, in the revised manuscript.

Point 2-       Line 84-86: Rephrase the product of co-transduction to enhance comprehension

Response 2: The product of co-transduction has been rephrased as followed: Dual CAR T-cell product can be produced by co-transduction of T cells with two different vectors, each of which encoding one individual CAR structure. It introduces pooled two separate Si-CAR-T and one Bi-CAR-T in the final product (Figure 1B, Co-transduction).

Point 3-       Line 96: Do all described tandem/loop CARs incorporate 41BB? Or are there also tandem/loop CARs with CD28 or other costimulatory domains?

Response 3: In the current review, most of the described tandem CARs incorporate 41BB except for the one targeting CD19/CD20 in the study by Zah (Cancer Immunol Res 2016), BCMA/CD19 by Kang (Biomark Res 2020), BCMA/CS1 by Zah (Nat Commun 2020). The described loop CARs with disclosed structure incorporate 41BB. However, the CAR structure of dual BCMA/CD19 targeted FasT CAR-T GC012F has not been disclosed. Whether the costimulatory domain of GC012F is CD28 or other costimulatory domains is unknown.

Point 4-       Line 102: Introduce bicistronic CAR before tandem and loop CAR structure because it’s more commonly used and better matches the figure

Response 4: Description of bicistronic CAR has been moved from Line 102 to Line 88 before tandem and loop CAR structure in the revised manuscript with simple markup.

Point 5-       Line 139: Transduction NOT transfection

Response 5: Transfected has been replaced by transduced.

Point 6-       Line 145: Product without s and delete products in line 147. The sentence should be “A pool of two Si-CAR-T products and one Bi-CAR-T product…”

Response 6: In Line 109-Line 110 of the revised manuscript with simple markup, the original sentence “(B), A pooled products of two Si-CAR-T products and one Bi-CAR-T product…” has been revised to “(B), A pool of two Si-CAR-T products and one Bi-CAR-T product…”. Please refer to the unlined words and the revised manuscript.

Point 7-       It should be discussed that tandem/loop CARs have one intracellular domain whereas bicistronic CARs have two intracellular T cell activating domains. Would that result in better activation of CARs when both antigens are presented? Are there any data investigating this hypothesis?

Response 7: There are two studies investigating the hypothesis that whether bicistronic CARs with two intracellular T cell activating domains results in better activation of CARs than tandem/loop CARs with one intracellular domain when both antigens are present.

In the study on dual CAR-T targeting BCMA/GPRC5D by de Larrea et al. (2020), a Bicistronic CAR (BCMA-4-1BB-GPRC5D-41BB) with two identical activating domains (two 41BBs), a Bicistronic CAR (BCMA-4-1BB-GPRC5D-CD28) with two different activating domains (one 41BB and the other CD28), and a Tandem CAR (GPRC5D-BCMA-4-1BB) with one activating domain (one 4-1BB) were compared. CAR-T transduced by the Bicistronic CAR with 4-1BB and CD28 showed limited efficacy. No activation data were presented in the study. In the schematic Figure 4 of the article by de Larrea et al. (2020), the Bicistronic CAR with two 41BBs is superior to the Tandem CAR with one 41BB; however, there was no significant difference between the two CARs from the cytotoxicity data and the animal survival data when both antigens were present. Therefore, it might need further analysis of data to draw conclusions on the activation.

In the study on dual CAR-T targeting CD5/CD7 by Dai et al. (2022), levels of CD107a, IFN-γ and TNF-α were comparable between CAR-T transduced with the Tandem CAR with one activating domain (41BB) and CAR-T with Bicistronic CAR with two activating domains (41BBs) when they were incubating with tumor cells with both antigens, indicating that the Tandem CAR with one activating domain might not be inferior to Bicistronic CAR with two activating domains in terms of activation.

Point 8-       Figure 2: Are these clinical trials registered at clinicaltrial.gov?

Response 8: Not all the clinical trials in Figure 2 are registered at clinicaltrials.gov. Trial registration can be found in the following table:

Author Year

Trial registration

Product

Tong 2020

NCT03097770

Tandem Bi-CAR-T targeting CD19/CD20

Shah 2020

NCT03019055

Tandem Bi-CAR-T targeting CD19/CD20

Dai 2020

NCT03185494

Tandem Bi-CAR-T targeting CD19/CD22

Wang 2020

ChiCTR-ORN-16008948; ChiCTR1800015575

Tandem Bi-CAR-T targeting CD19/CD22

Schultz 2019

Unavailable

Tandem Bi-CAR-T targeting CD19/CD22

Mei 2021

ChiCTR1800018143

Tandem Bi-CAR-T targeting BCMA and CD38

Pan 2020

ChiCTR-OIB-17013670

cocktail/sequential infusion of two separate Si-CAR-T products targeting CD19/CD22

Wang 2020

ChiCTR-OPN-16008526

cocktail/sequential infusion of two separate Si-CAR-T products targeting CD19/CD22

Tang 2020

Unavailable

cocktail/sequential infusion of two separate Si-CAR-T products targeting BCMA/CD19

Yan 2019

ChiCTR-OIC-17011272

cocktail/sequential infusion of two separate Si-CAR-T products targeting BCMA/CD19

Yang 2020

NCT04129099

Loop Bi-CAR-T targeting CD19/CD22

Zhang 2021

NCT03196830

Loop Bi-CAR-T targeting CD19/CD22

Spiegel 2021

NCT03233854

Loop Bi-CAR-T targeting CD19/CD22

Tholouli 2020

NCT03287817; EudraCT 2016-004682-11

Bicistronic Bi-CAR-T targeting CD19/CD22

Cordoba 2021

NCT03289455; EUDRA CT 2016-004680-39

Bicistronic Bi-CAR-T targeting CD19/CD22

Gardner 2020

NCT03330691

CD19/CD22 Bi-CAR-T produced by co-transduction

Yang 2018

Unavailable

CD19/CD22 Bi-CAR-T produced by co-transduction

Point 9-       Figure 2: Should be included later in the manuscript when the clinical trials are discussed

Response 9: Agree. Figure 2 was intended to be inserted in Section 4 (Comparison of different…) of the original manuscript submitted by authors. Please refer to Line 439 in the revised manuscript with simple markup.

Point 10-       Line 174-176: Please add short explanations for OR-gate, AND, NOT and synNotch strategy

Response 10: short explanations for OR-gate, AND, NOT and synNotch strategy have been added in the round brackets as underlined below and the revised manuscript:

Of note, two pooled Si-CAR-T products, and Bi-CAR-T, are sometimes referred to as OR-gate CAR-T or CAR-T using “OR” logic gate (activated by one antigen or the other on tumor cells). CAR-T cells using other logic gates such as “AND” (only activated when recognizing both antigens on tumor cells), “NOT” (inactivated when encountering one antigen on normal cells), and “synNotch” (first primed and induced by one tumor-specific but heterogeneous antigen on tumor cells and then activated by one homogeneous antigen or the other homogeneous antigen on tumor cells) are also being developed, predominantly for solid tumors.

Point 11-       Line 191: Transduction not transfection

Response 11: Transfection has been corrected to transduction.

Point 12-       Line 265: The authors should consider adding trogocytosis as a mechanism of CAR T failure

Response 12: Thank you for providing the novel mechanism trogocytosis.

The sentence “Currently, investigations reveal that there are three main mechanisms responsible for relapse due to antigen loss: receptor genetic mutations, cell lineage switch, and epitope masking” has been revised as below:

“Currently, investigations reveal that there are four main mechanisms responsible for relapse due to antigen escape: receptor genetic mutations, cell lineage switch, epitope masking and trogocytosis”

And the brief explanation of trogocytosis is added at the end of the paragraph from Line 234 of the revised manuscript with simple markup as below:

“In recent years, tumor cells are found to be able to transfer the target antigen to CAR-T via trogocytosis, resulting in diminished antigen expression on tumor cells and fratricide of CAR-T”.

Point 13-       Line 276-78: Please rephrase this sentence

Response 13: The sentence “Recently, Ruella et al. (2018) reported a rare case of epitope masking by binding unexpected CARs to B cells to the CD19 epitope, thus blocking the binding of CD19 Si-CAR-T to malignant B cells” has been rephrased as “Ruella et al. (2018) reported a rare case of epitope masking caused by unintentionally transducing B cells with CAR against CD19; the expression of CAR on the resulting CAR-transduced B cell leukemia cells (CARB) bound to the CD19 epitope of the same CARB, thus blocking the binding of CD19 Si-CAR-T to CARB” in the Line 228 of the revised manuscript with simple markup.

Point 14-       Line 294-296: Are these ALL patients that have been treated with CART19?

Response 14: Yes. The ALL patients from the sentence “The time to CD19-negative or CD19-dim relapse was reported to be around 2–3 months in five patients, 4–6 months in four patients, 8–9 months in five patients, and 14 months in one patient” (sentence A) have been treated with CART19. The data from the sentence A were summarized from the following two tables:

Sotillo et al. Convergence of Acquired Mutations and Alternative Splicing of CD19 Enables Resistance to CART-19 Immunotherapy. Cancer Discov. 2015 Dec;5(12):1282-95. doi: 10.1158/2159-8290.CD-15-1020.

Orlando et al. Genetic mechanisms of target antigen loss in CAR19 therapy of acute lymphoblastic leukemia. Nat Med. 2018 Oct;24(10):1504-1506. doi: 10.1038/s41591-018-0146-z.

Point 15-       Line 341-343: Please rephrase this sentence to improve comprehension

Response 15: The sentence “For example, OS for ALL patients were CD19 Si-CAR-T ~ (close to) CD22 Si-CAR-T < CD19/CD22 Bi-CAR-T < cocktail infusion of CD19 or CD22 Si-CAR-T” has been rephrased to “For example, OS for ALL patients treated with CD19 Si-CAR-T were close to those with CD22 Si-CAR-T, which were shorter than those treated with CD19/CD22 Bi-CAR-T. OS for ALL patients treated with cocktail infusion of CD19 or CD22 Si-CAR-T was the longest among those with Si-CAR-T, Bi-CAR-T and cocktail infusion of two Si-CAR-T”.

Point 16-       Line 378-379: Please rephrase this sentence to improve comprehension

Response 16: The sentence “Overcoming BCMA-negative or BCMA-low escape has been proposed as capable of reversing resistance to BCMA Si-CAR-T” has been rephrased to “Overcoming BCMA-negative or BCMA-low escape has been proposed as capable of reversing resistance of malignant plasma cells to BCMA Si-CAR-T”.

Reviewer 2 Report

This is a review work about the potential of bispecific CAR-Ts in treatment of lymphoid malignancies. Though it is clear that authors really put significant effort to this paper and summarized a lot of knowledge, the material is not well organized and therefore, readability of the article is quite challenging. Tables are busy and not well designed from the graphical point of view.  Also, the language would benefit from some improvement (as long as non-native speaker is able to evaluate). 

Recommendation regarding tables:

1. Make tables as wide as possible and appropriate to ensure that information in one cell do not have too many rows. In Cancers, it is possible to use the landscape page design, which would be benefit especially Table 2 and 3. Divide table rows by lines. It may be also possible to use smaller lettering. 

2. Put in the tables only content that is appropriate/necessary. For example, in Table 2 I am not sure if all details of the construct (including linker and hinge segments, which relevance is not discussed elsewhere in the text) need to be included. 

3. For the references in tables, use only first name of author and citation number. In Table 2, put the referenced articles in the first column. 

Other major points:

1. This paper is targeted both on preclinical and clinical researchers, but the preclinic is not always well distinguished from the clinics. 

2. In discussion of various CAR designs, mention the possibility of non-viral transduction. Transposones enable packaging of larger gene constructs. 

3. When reporting efficacy of CD19/CD20 and CD19/CD22 CAR-Ts, be sure to distinguish ALL from NHL. This applies both to efficacy and toxicity, which may differ in these diseases. Even for NHL, it is unfair to draw conclusions from trials aimed at different clinical entities (Lines 316-318). Also, the problem with CD19 negativity is different in ALL and NHL. In ALL, the CD19 negativity is easy to evaluate by flow cytometry. In NHL, which are not usually leukemised, we rely on tissue biopsy, which is not always repeated on relapse. If it is repeated, immunohistochemical staining for CD19 is less sensitive than flow cytometry. 

5. For earlier phase studies, the PFS/OS are not the best parameters to be evaluated, as patients may be consolidated with allogeneic transplant even before CAR-T failure, or after failure, they may receive other therapies. Overall and complete rate responses are more appropriate. Conclusions about efficacies of different approaches are not fair, as they do not take in account possible differences in patients characteristics and wide confidence intervals for PFS/OS especially in smaller studies. 

6. For cytokine release syndome, there are several scoring systems and therefore, results of different studies may not be comparable. I have already mentioned that toxicities may differ in ALL and NHL with the same product, but they also differ according to costimulatory molecules used (CD28 v. 4-1BB). This may be more significant than other differences in CAR design. The differences in management of CRS in earlier and later studies are already discussed.

Minor point: 

In the Section 2 (Common dual CAR strategies) I suggest to list bicistronic Bi-CAR-Ts as third approach, to correspond with the Figure 1.

Author Response

Response to Reviewer 2 Comments

Point 1: Recommendation regarding tables:

Point 1.1. Make tables as wide as possible and appropriate to ensure that information in one cell do not have too many rows. In Cancers, it is possible to use the landscape page design, which would be benefit especially Table 2 and 3. Divide table rows by lines. It may be also possible to use smaller lettering.

Response 1.1: Thank you for the advice. Please refer to the reformatting tables in the revised manuscript.

1.2. Put in the tables only content that is appropriate/necessary. For example, in Table 2 I am not sure if all details of the construct (including linker and hinge segments, which relevance is not discussed elsewhere in the text) need to be included.

Response 1.2: Thank you for the advice. Linkers have been discussed from Line 167 in section 2 (Common dual CAR strategies) in the revised manuscript with simple markup. The number of amino acids for hinge segments are listed in the table. It’s related to the size of the CAR. The abbreviation of aa is added to Table 2.

  1. For the references in tables, use only first name of author and citation number. In Table 2, put the referenced articles in the first column.

Response 1.3: Thank you for the advice. The references in tables have been revised.

Point 2: Other major points:

Point 2.1. This paper is targeted both on preclinical and clinical researchers, but the preclinic is not always well distinguished from the clinics.

Response 2.1: Does the referee refer to the preclinic data that is not as good as the clinical data? The preclinical data were summarized from published papers by different research groups. It may not be as practical as the clinical data. It is difficult to conduct head-to-head comparisons in the clinical setting.

  1. In discussion of various CAR designs, mention the possibility of non-viral transduction. Transposones enable packaging of larger gene constructs.

Response 2.2: Thank you for the idea. The sentence “Non-viral transduction using transposons for producing Bi-CAR is possible to increase transduction efficiencies owing to the capacity of transferring large gene constructs” with a review on transposons (Tipanee, 2017) has been added and cited at the end of the paragraph starting with “Some features of Bi-CAR are related to druggability” in Line 165 of the revised manuscript with simple markup. Please recommend other references to be cited.

  1. When reporting efficacy of CD19/CD20 and CD19/CD22 CAR-Ts, be sure to distinguish ALL from NHL. This applies both to efficacy and toxicity, which may differ in these diseases. Even for NHL, it is unfair to draw conclusions from trials aimed at different clinical entities (Lines 316-318). Also, the problem with CD19 negativity is different in ALL and NHL. In ALL, the CD19 negativity is easy to evaluate by flow cytometry. In NHL, which are not usually leukemised, we rely on tissue biopsy, which is not always repeated on relapse. If it is repeated, immunohistochemical staining for CD19 is less sensitive than flow cytometry.

Response 2.3: We agree that efficacy and toxicity differ between ALL and NHL and therefore we separated tables of ALL from tables of NHL to summarize efficacy and toxicity data with separate descriptions in the main text. The sentence “Likewise, the 12-month PFS was close among CD19 Si-CAR-T tisagenlecleucel, brexucabtagene autoleucel, and CD19/CD20 tandem Bi-CAR-T in NHL” has been revised to “Likewise, the 12-month PFS for NHL patients was close among CD19 Si-CAR-T tisagenlecleucel in diffuse large B-cell lymphoma, CD19 Si-CAR-T brexucabtagene autoleucel in mantle-cell lymphoma, and CD19/CD20 tandem Bi-CAR-T in B-cell lymphoma; however, the comparison should be viewed with cautions at different clinical entities.”  

We also agree with the difference of CD19 negativity between ALL and NHL. It’s mentioned in the review by Majzner and Mackall cited by our manuscript.

  1. For earlier phase studies, the PFS/OS are not the best parameters to be evaluated, as patients may be consolidated with allogeneic transplant even before CAR-T failure, or after failure, they may receive other therapies. Overall and complete rate responses are more appropriate. Conclusions about efficacies of different approaches are not fair, as they do not take in account possible differences in patients characteristics and wide confidence intervals for PFS/OS especially in smaller studies.

Response 2.5: Thank you for pointing out the inadequacies. We have added a paragraph to include the points as limitations in the last section (5 Challenges and perspectives).

  1. For cytokine release syndome, there are several scoring systems and therefore, results of different studies may not be comparable. I have already mentioned that toxicities may differ in ALL and NHL with the same product, but they also differ according to costimulatory molecules used (CD28 v. 4-1BB). This may be more significant than other differences in CAR design. The differences in management of CRS in earlier and later studies are already discussed.

Response 2.6: Thank you for pointing out the inadequacies. We have added a paragraph to include the point as limitation in the last section (5 Challenges and perspectives).

Point 3: Minor point:

Point 3.1 In the Section 2 (Common dual CAR strategies) I suggest to list bicistronic Bi-CAR-Ts as third approach, to correspond with the Figure 1.

Response 3.1: Thank you for the suggestion. Bicistronic Bi-CAR-Ts has been listed as third approach.

Round 2

Reviewer 2 Report

The revised version presents improvement from that presented originally. Ordering of different Bi-CAR-T strategies is now the same as in Figure 1. Orientation in Tables is much better than in previous version. I also appreciate the clarification of issues connected with preclinical/clinical data. Regarding the point 2.1 and authors´ answer to it: I did not imply preclinical data are inferior, just that these two types of data should be discussed separately.

The only thing is the added paragraph to Challenges and perspectives (row 616-24): In an article, I would not use such strong formulations as I did in the Review. I suggest following adjustment:

The limitations of the review are as followed: (1) For earlier phase studies, PFS/OS estimations are based on small numbers of patients. Also, it should be taken into account that patients may receive additional treatment after CAR-T; (2) Comparison of efficacies of different approaches must take into account wide confidence intervals of PFS/OS estimation, (3) Comparison of safety profiles must take into account differences in scoring systems for toxicity and changes in toxicity management over time. 

Author Response

Response to Reviewer 2 Comments

Point 1: Regarding the point 2.1 and authors´ answer to it: I did not imply preclinical data are inferior, just that these two types of data should be discussed separately.

Response 1: We agree that it would be better to discuss preclinical data and clinical data separately. However, because it is financially difficult for CAR-T to conduct head-to-head comparisons in the clinical setting, we had to discuss the preclinical data with the clinical data.

Point 2: The only thing is the added paragraph to Challenges and perspectives (row 616-24): In an article, I would not use such strong formulations as I did in the Review. I suggest following adjustment:

The limitations of the review are as followed: (1) For earlier phase studies, PFS/OS estimations are based on small numbers of patients. Also, it should be taken into account that patients may receive additional treatment after CAR-T; (2) Comparison of efficacies of different approaches must take into account wide confidence intervals of PFS/OS estimation, (3) Comparison of safety profiles must take into account differences in scoring systems for toxicity and changes in toxicity management over time.

Response 2: Thank you for revising the paragraph. The adjustment has been made in the section of Challenges and perspectives from Line 547-553 in the revised manuscript with simple markup.